# The Impacts of International Political and Economic Events on Japanese Financial Markets

**Mirzosaid Sultonov** 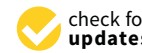

Department of Community Service and Science, Tohoku University of Community Service and Science, Sakata 9988580, Japan; sultonov@koeki-u.ac.jp; Tel.: +81-234-41-1255

**Abstract:** Information about the possibilities of changes in national and international macroeconomic variables affects the expectations and behavior of individuals and firms more quickly than real changes in those macroeconomic variables. In this research, we investigate the impacts of international information (news) on the financial markets in Japan. We examine how news about the results of the Brexit referendum (BR) and the United States presidential election (USE) affected foreign exchange rates and stock market indexes. This research reveals evidence of statistically significant changes in exchange rates and stock market indexes within two weeks after the BR and USE, statistically significant changes in the exchange rate variance within the first week after the BR, and changes in the causality relationship between the variables after each event.

**Keywords:** Japan; financial markets; impact of news; Brexit referendum; United States presidential election

## 1. Introduction

In developed countries such as Japan, financial markets, particularly foreign exchange and stock markets, are highly sensitive to changes in international economic and political environments. The Japanese foreign exchange and stock markets immediately respond to every event with significant political and economic consequences. Changes in exchange rates and stock market indexes have impacts on major economic fundamentals, and research on variations in the exchange rate and stock market indexes is important.

Economic theory provides the reader with many theories and models that define the impacts of different factors (both national and international) on domestic macroeconomic variables. Events with unexpected results such as the Brexit referendum (BR) and the United States presidential election (USE), affect trade policy, which impacts the real trade volume and economies of trade partners such as Japan. This process can take many months. However, in the first days after the results of these specific events were announced, daily exchange rates and stock market indexes changed significantly in Japan and many other countries around the world. Such quick responses by financial markets are related to the impacts of information and news. Both individuals and professional investors have different expectations about the immediate effects of events and change their demand and supply soon after reading and viewing information and news about important domestic and international events.

Different types of generalized autoregressive conditionally heteroscedasticity (GARCH) models are used to analyze the impacts of a variety of national and international factors on financial markets. In particular, the impacts of international economic and financial changes (Chung and Jang 2000; Agren 2006; Karfakis and Panagiotidis 2015), natural disasters (Hanabusa 2010; Wang and Kutan 2013),

and political changes and conflicts (Lin and Wang 2005; Hanabusa 2010) on the financial markets of Japan have been studied.

The impacts of the BR and USE and the policy uncertainty induced by both events on financial markets has attracted extensive attention. According to Shaikh (2017), the USE had profound effects on the global financial markets, and the effects have persisted in equity and foreign exchange markets across the globe. According to Belke et al. (2018), Brexit-induced policy uncertainty caused instability in financial markets and had the potential to damage the economies in the United Kingdom (UK) and other European countries. According to Bashir et al. (2019), the results of the Brexit referendum led to a notable shift in financial markets and caused positive and negative comovements in the UK and European Union financial markets. In their investigation of the impacts of Brexit-related events on the corporate bond yield spreads in the UK and Eurozone, Kadiric and Korus (2019) indicated that the announcement of the referendum results was associated with increasing credit spreads.

In our previous research article (Sultonov and Jehan 2018), which used a dynamic conditional correlation bivariate GARCH model[1] and one-sample *t*-test, we analyzed the impact of the BR and USE on the dynamic conditional correlation between the exchange rate and stock price index in Japan. The empirical findings showed a significant change in the dynamic conditional correlation coefficients after each event. In the current paper, we examined the impact of information about the BR and USE on the returns and volatility of the exchange rate and stock price index in Japan, the asymmetry of the news impact on the volatility of the exchange rate and stock price index, and changes in the causality relationship between the exchange rate and stock price index.

The next section briefly explains the methodology. Section 3 describes the data, and Section 4 presents the results of the empirical analysis. The last section concludes the paper.

## 2. Methodology

According to Ross (1989), an analysis of return volatility provides useful data on the information flow between markets. Bollerslev et al. (1992), referring to a huge number of empirical studies, demonstrated that autoregressive conditional heteroscedasticity (ARCH) models were very suitable for modeling the time-varying volatility of financial data.

ARCH models estimate future volatility as a function of prior volatility and are widely used in the analysis of the volatility of financial data. The fundamental ARCH model suggested by Engle (1982) was further developed for the generalized ARCH (GARCH) by Bollerslev (1986), the absolute error model by Taylor (1986) and Schwert (1989), the exponential GARCH (EGARCH) by Nelson (1991), and the GJR (Glosten, Jagannathan, and Runkle) model by Glosten et al. (1993).

In the EGARCH model, as the log value of volatility is used as an explained variable, there is no need to impose non-negative constraints on model parameters (Hamori 2003). Furthermore, the EGARCH model considers the asymmetric effect of volatility, and through this model, we can observe both the symmetric and asymmetric impacts of information. These features of the model enable us to easily incorporate dummy variables for each event into mean and variance equations and obtain more detailed information about the impact of international events on the return and volatility of the exchange rate and stock price index.

In our study, we applied the EGARCH model (Nelson 1991) to calculate the conditional mean and conditional variance. The conditional mean equation can be written as

$$r_t = C + \sum_{i=1}^{k} a_i r_{t-i} + \varepsilon_t. \tag{1}$$

---

[1] The model was developed by Engle and Sheppard (2001) and Engle (2002).

The variance equation can be written as

$$ln(\sigma_t^2) = \omega + \sum_{i=1}^{p}(\gamma_i \varepsilon_{t-i}/\sigma_{t-i} + \alpha_i(|\varepsilon_{t-i}/\sigma_{t-i}| - (2/\pi)^{1/2})) + \sum_{i=1}^{q}\beta_i ln(\sigma_{t-i}^2). \tag{2}$$

In Equation (1), the returns ($r$) at time $t$ are the function of a constant ($C$), previous returns, and information ($\varepsilon$) available at time $t$. In Equation (2), the variance ($\sigma^2$) at time $t$ is the function of a constant ($\omega$), past variance, and past news about volatility. Parameter $\alpha$ in Equation (1) is the coefficient for the impacts of previous returns. Parameters $\gamma$, $\alpha$, and $\beta$ in Equation (2) are the coefficients for the asymmetric impacts of past news, symmetric impacts of past news, and impacts of past variance, respectively. A negative sign for these parameters is not precluded. Dummy variables for the first week after the BR and USE and for the first and second weeks combined after the BR and USE are incorporated as additional independent variables in the conditional mean and variance equations. Parameters $k$, $p$ and $q$ in Equations (1) and (2) are specified based on the Schwarz–Bayesian information criterion (SBIC) and the log-likelihood ratio. The Ljung–Box $Q$ test is used to evaluate the robustness of the model specification.

To examine the changes in the causality relationship between the exchange rate and stock price index, we followed the procedures proposed by Hong's (2001). We computed the sample cross-correlation function $\hat{\rho}_{uv}(j)$ between the standardized residuals and squared standardized residuals derived from the estimations for Equations (1) and (2). After that, we chose a weighting function $k(z)$ and an integer $M$ and computed the components $C_{1T}(k)$ and $D_{1T}(k)$ used in the estimation of Hong's (2001) $Q$-statistics. The computed $Q$-statistics were compared to the upper-tailed critical value of $N(0;1)$ at an appropriate level. A $Q$-statistics value larger than the critical value means the null hypothesis of "no causality" is rejected.

The sample cross-correlation coefficient $\hat{\rho}_{uv}(j)$ at lag $j$ is estimated as

$$\hat{p}_{uv}(j) = c_{uv}(j)(c_{uu}(0)c_{vv}(0))^{-1/2}, \tag{3}$$

where $c_{uv}(j)$ is the $j$-th lag sample cross covariance and $c_{uu}(0)$ and $c_{vv}(0)$ are the sample variances of $u$ and $v$. Here, $u$ and $v$ are the standardized residuals derived from Equations (1) and (2) for the exchange rate and stock price index. Standardized residuals are used when estimating causality in mean, and squared standardized residuals are used instead of standardized residuals when estimating causality in variance.

Hong's (2001) $Q$-statistic is computed as

$$Q = \left\{ T\sum_{j=1}^{T-1}k^2(j/M)\hat{\rho}_{uv}(j) - C_{1T}(k) \right\} /\{2D_{1T}(k)\}^{1/2}, \tag{4}$$

where

$$C_{1T}(k) = \sum_{j=1}^{T-1}(1 - j/T)k^2(j/M) \tag{5}$$

and

$$D_{1T}(k) = \sum_{j=1}^{T-1}(1 - j/T)\{1 - (j+1)/T\}k^4(j/M). \tag{6}$$

To compute Hong's (2001) $Q$-statistics, we chose the weighting function $k(z)$. We selected the truncated kernel as

$$k(z) = \begin{cases} 1, & |z| \leq 1, \\ 0, & otherwise. \end{cases}$$

The asymptotic power of Hong's (2001) *Q*-statistics depends on $k(z)$, and the truncated kernel gives equal weight to each lag at sample cross correlations.

## 3. Data Description

In the estimations, we used the logarithmic returns series of the daily exchange rates for the Japanese Yen (JPY) against the US dollar and the closing price indexes of the Nikkei and Tokyo Stock Price Index (TOPIX) from 9 February 2016 to 24 March 2017. This period covered a total of 276 observations including an equal number of observations before and after the BR and USE. These time series came from the Bank of Japan and Google Finance. Weekends and holidays were excluded.

Table 1 presents the descriptive statistics for the returns. The mean values were very close to zero. The standard deviation values showed relatively higher instability in the stock price index returns. Skewness, kurtosis, and the Jarque–Bera test (Jarque and Bera 1987) indicated that the returns were not normally distributed. The augmented Dickey–Fuller (ADF) unit root test (Dickey and Fuller 1979, 1981) proved that the return series were stationary.

**Table 1.** Daily logarithmic returns of the exchange rates and stock price indexes.

| Variables | Obs. | Mean | Std. Dev. | Skewness | Kurtosis | Jarque–Bera | ADF |
|---|---|---|---|---|---|---|---|
| | | Whole period: 9 February 2016 to 24 March 2017 | | | | | |
| JPY | 276 | −0.00019 | 0.00840 | −0.54315 | 8.59034 | 373.00 *** | −3.633 *** |
| Nikkei | 276 | 0.00045 | 0.01514 | −0.45002 | 9.52024 | 498.20 *** | −5.304 *** |
| TOPIX | 276 | 0.00041 | 0.01458 | −0.36411 | 9.87909 | 550.30 *** | −5.303 *** |
| | | Before BR: 9 February 2016 to 23 June 2016 | | | | | |
| JPY | 92 | −0.00123 | 0.00865 | −1.95794 | 11.7919 | 355.10 *** | −3.468 ** |
| Nikkei | 92 | −0.00050 | 0.01856 | 0.05303 | 5.24109 | 19.300 *** | −3.264 ** |
| TOPIX | 92 | −0.00066 | 0.01866 | 0.14395 | 6.35728 | 43.520 *** | −3.068 *** |
| | | After BR and before USE: 24 June 2016 to 8 November 2016 | | | | | |
| JPY | 92 | −0.00002 | 0.00793 | −0.61696 | 5.09002 | 22.580 *** | −3.467 ** |
| Nikkei | 92 | 0.00061 | 0.01402 | −2.01021 | 15.1419 | 627.10 *** | −3.381 *** |
| TOPIX | 92 | 0.00053 | 0.01324 | −1.82225 | 13.5368 | 476.50 *** | −3.347 *** |
| | | After USE: 9 November 2016 to 24 March 2017 | | | | | |
| JPY | 92 | 0.00069 | 0.00857 | 0.96898 | 6.62240 | 64.700 *** | −2.959 ** |
| Nikkei | 92 | 0.00125 | 0.01224 | 0.54557 | 13.9792 | 466.60 *** | −2.951 ** |
| TOPIX | 92 | 0.00135 | 0.01081 | 0.46879 | 12.6791 | 362.50 *** | −2.657 * |

Note: *** in the Jarque–Bera test results indicates that the null hypothesis of normal distribution is rejected at the 1% significance level. The maximum number of lags for the augmented Dickey–Fuller (ADF) test selected by the Schwarz–Bayesian information criterion (SBIC) was 15 for the whole period and 11 for the subperiods. For the ADF test, ***, ** and * indicate values less than the critical value at 1%, 5%, and 10% significance levels.

Figure 1 illustrates the levels and logarithmic returns series of the exchange rates and stock price indexes. The vertical reference lines indicate the days in which the BR and USE took place. In the figure, we can observe changes in the levels and returns after each event.

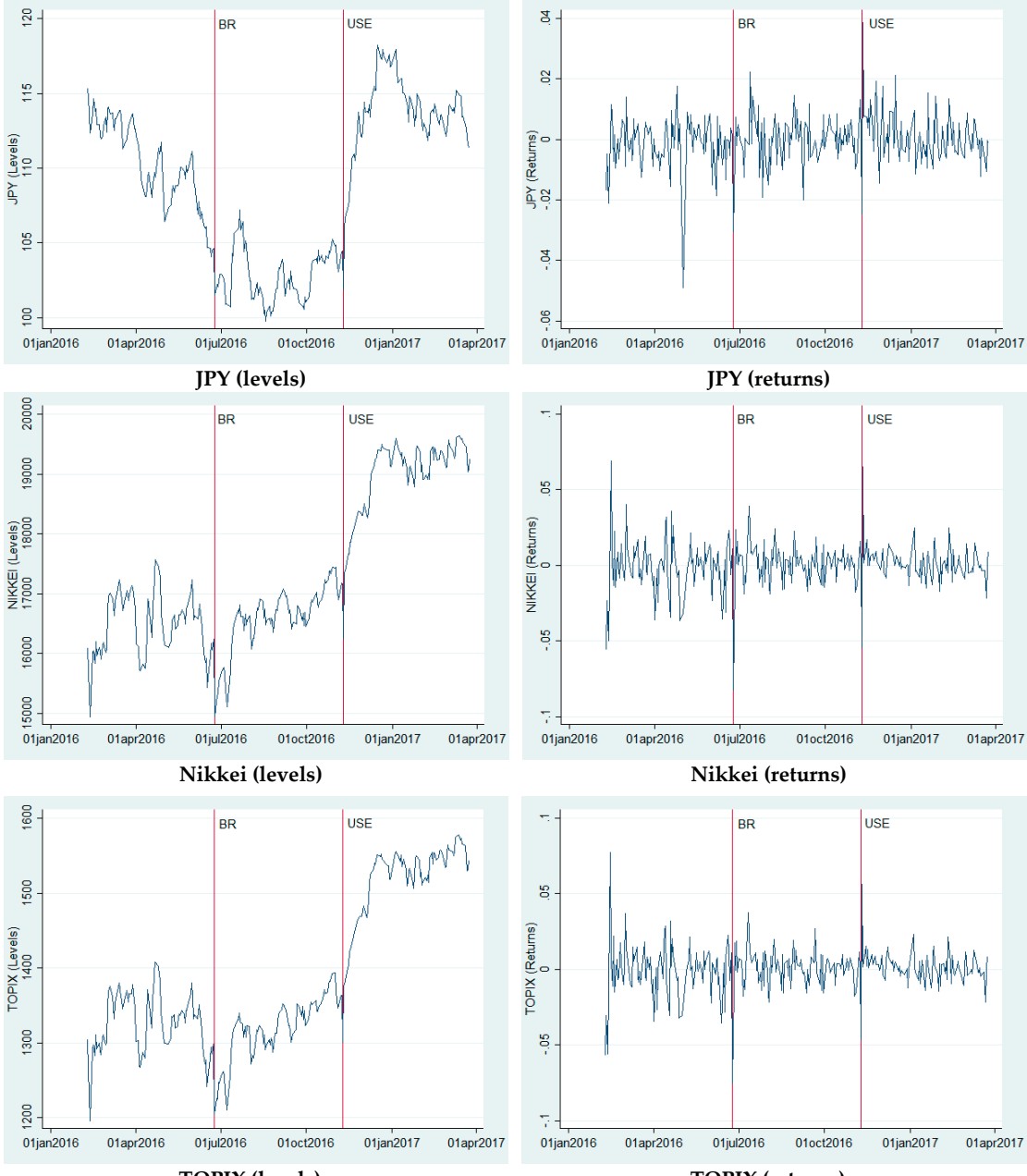

**Figure 1.** Levels and returns for the exchange rates and stock price indexes.

## 4. Empirical Results

### 4.1. Model Estimations

Table 2 presents the estimation results of the autoregressive (AR-) EGARCH model for each variable. The number of lags was specified based on the SBIC and residuals diagnostic. The first equation was estimated without a dummy. The other equations incorporated dummy variables for the first week and then the two weeks after the BR and USE. The dummy variables were included as independent variables in the conditional mean and variance equations. The dummy variables took the value of 1 for observations in the first week and then the two weeks after the BR and USE and the value of 0 for all other observations.

**Table 2.** Model estimations.

| $r_t$ | JPY | JPY | JPY | Nikkei | Nikkei | Nikkei | TOPIX | TOPIX | TOPIX |
|---|---|---|---|---|---|---|---|---|---|
| **Mean** | | | | | | | | | |
| $C$ | 0.0000 | −0.0001 | −0.0003 *** | 0.0000 | 0.0004 ** | 0.0001 *** | 0.0004 *** | 0.0002 | 0.0004 |
| | (0.0005) | (0.0029) | (0.0001) | (0.0002) | (0.0002) | (0.0000) | (0.0001) | (0.0009) | (0.0008) |
| $r_{t-1}$ | −0.0205 | −0.0430 | −0.0940 *** | 0.0265 *** | 0.0373 | 0.0086 ** | 0.0297 ** | 0.1099 | 0.0247 |
| | (0.0869) | (0.3057) | (0.0022) | (0.0028) | (0.1296) | (0.0038) | (0.0137) | (0.1164) | (0.0782) |
| Brexit 1 week | | 0.0035 | | | −0.0017 | | | −0.0333 | |
| | | (0.0043) | | | (0.0430) | | | (0.0373) | |
| Brexit 2 weeks | | | −0.0004 *** | | | −0.0062 | | | −0.0048 *** |
| | | | (0.0000) | | | (0.0081) | | | (0.0009) |
| USE 1 week | | 0.0084 | | | −0.0013 | | | −0.0066 | |
| | | (0.0107) | | | (0.0062) | | | (0.0180) | |
| USE 2 weeks | | | 0.0084 *** | | | 0.0030 *** | | | 0.0028 |
| | | | (0.0002) | | | (0.0010) | | | (0.0029) |
| **Variance** | | | | | | | | | |
| $\omega$ | −17.037 *** | −17.744 *** | −19.152 *** | −0.4111 ** | −6.7280 | −0.5606 | −0.4377 * | −0.8037 | −0.4830 |
| | (2.8158) | (1.1457) | (1.2575) | (0.1937) | (7.6965) | (0.3791) | (0.2253) | (0.5124) | (0.9977) |
| $\gamma_1$ | −0. 1143 | −0.1253 | −0.0470 | −0.3005 *** | −0.3462 * | −0.3002 *** | −0.2934 *** | −0.3987 *** | −0.2919 *** |
| | (0.0727) | (0.0792) | (0.0829) | (0.0746) | (0.1861) | (0.0794) | (0.0710) | (0.1260) | (0.1085) |
| $\alpha_1$ | 0.0289 | −0.0197 | −0.0127 | 0.0303 | 0.3277 | 0.0912 | 0.0653 | 0.1312 | 0.0950 |
| | (0.1806) | (0.1490) | (0.1146) | (0.0550) | (0.3088) | (0.1093) | (0.0662) | (0.1272) | (0.3392) |
| $\beta_1$ | −0.7649 *** | −0.8206 *** | −0.9651 *** | 0.9532 *** | 0.2360 | 0.9350 *** | 0.9517 *** | 0.9111 *** | 0.9458 *** |
| | (0.2939) | (0.1229) | (0.1084) | (0.0221) | (0.8813) | (0.0442) | (0.0251) | (0.0575) | (0.1149) |
| Brexit 1 week | | 2.1566 * | | | 2.0436 | | | 0.5064 | |
| | | (1.2185) | | | (1.4497) | | | (0.8815) | |
| Brexit 2 weeks | | | 1.3502 | | | 0.0598 | | | 0.0160 |
| | | | (1.1926) | | | (0.1573) | | | (0.0819) |
| USE 1 week | | 1.0209 | | | 1.5501 | | | 0.0544 | |
| | | (1.2260) | | | (1.9073) | | | (0.6265) | |
| USE 2 weeks | | | 0.9928 | | | −0.2012 | | | −0.1171 |
| | | | (0.9905) | | | (0.1998) | | | (0.5512) |
| GED parameter | 0. 1204 | 0.1668 | 0.1608 | 0.1253 | 0.1980 | 0.1400 | 0.1928 | 0.4538 * | 0.1943 |
| | (0.1291) | (0.1543) | (0.1334) | (0.1215) | (0.3048) | (0.1178) | (0.1280) | (0.2597) | (0.1466) |
| **Diagnostic** | | | | | | | | | |
| $Q(5)$ | 5.0193 | 3.8026 | 3.5284 | 2.5675 | 6.7945 | 1.7072 | 3.0440 | 3.5301 | 2.7364 |
| | (0.4135) | (0.5782) | (0.6191) | (0.7663) | (0.2364) | (0.8880) | (0.6932) | (0.6188) | (0.7405) |
| $Q^2(5)$ | 1.0372 | 1.2148 | 0.9833 | 1.4679 | 6.2153 | 1.2978 | 1.6036 | 3.9060 | 2.0401 |
| | (0.9595) | (0.9434) | (0.9639) | (0.9167) | (0.2858) | (0.9352) | (0.9008) | (0.5630) | (0.8436) |

Note: The numbers in parentheses are standard errors. $Q$ (5) is the Ljung–Box $Q$-statistics for the null hypothesis of no autocorrelation of up to order 5 for the standardized residuals. ***, ** and * indicate significance at 1%, 5%, and 10% levels.

In the mean equation, the coefficients of the previous day's returns were negative for the exchange rates and positive for the stock price indexes. In the equation without the dummy variables, the impacts were negative and statistically insignificant for the exchange rates and positive and statistically significant (at 1–5% significance levels) for the stock price indexes. Incorporating the dummy variables changed the value and power of the coefficients of the previous day's returns. For exchange rates or stock price indexes, the previous day's returns had no statistically significant impacts on the current day's returns in the equation with dummy variables for a week. For the exchange rates, the previous day's returns had negative and statistically significant impacts (at a 1% significance level) on the current day's returns in the equation with dummy variables for two weeks. In the equation with dummy variables for two weeks, the impacts of the previous day's returns were positive and statistically significant (at a 5% significance level) for the Nikkei, but insignificant for the TOPIX.

In the mean equation with dummy variables for a week, the coefficients of the dummy variables were positive for exchange rates and negative for stock price indexes. The impacts were statistically insignificant. The coefficients of the dummy variables for two weeks were negative for the BR and positive for the USE. The impacts of the BR dummy were statistically significant for exchange rates and the TOPIX (at a 1% significance level). The impacts of the USE dummy were statistically significant for the exchange rates and the Nikkei (at a 1% significance level). These significant impacts mean that the exchange rates appreciated and the stock price index (TOPIX) decreased in the first two weeks after the BR, while the exchange rates depreciated and the stock price index (Nikkei) increased in the first two weeks after the USE.

In the equation for the conditional variance, the coefficients for the asymmetric impacts of information had a negative sign, which means that negative news has more impact than positive news on the variance in all the equations. Furthermore, the absolute value of the coefficients for the asymmetric impacts of information was larger than the absolute value of the coefficients for the symmetric impacts of information. The asymmetric impact coefficients were statistically significant (at 1–10% significance levels) for the variance of stock price indexes. The symmetric impact coefficients were statistically insignificant for all equations; the signs were positive in all equations, except the equations for exchange rates with dummy variables. The coefficients of the previous period's variance were negative for exchange rates (in the EGARCH model, negative values for parameters are not precluded) and positive for the variance of stock price indexes. The impacts were statistically significant for all equations, except for equations with dummy variables for one week in the case of the Nikkei. These results mean that previous variance has significant impacts on conditional variance.

The coefficients of BR and USE dummy variables for one week were positive for all variables, but the impacts had weak significance (at a 10% significance level) only for exchange rates. These results mean that exchange-rate volatility increased in the first week after the BR. The coefficients of BR dummy variables for two weeks were positive and statistically insignificant for all variables. The coefficients of USE dummy variables for two weeks were positive and statistically insignificant for exchange rates and negative and statistically insignificant for stock price indexes.

The portmanteau test of white noise developed by Box and Pierce (1970) and refined by Ljung and Box (1978) does not reject the null hypothesis that there is no autocorrelation up to order 5 for standardized residuals and the squared values of standardized residuals. The test supports the robustness of the model specification.

### 4.2. Changes in Causality Relationship

The changes in returns and volatility of exchange rates and stock price indexes caused by the BR and USE might affect the causality relationship between the exchange rate and stock price indexes.

Table 3 presents the coefficients of the cross-correlation function with the maximum absolute value of Hong's (2001) *Q*-statistics within the first 10 lags for standardized residuals. The coefficients showed a unidirectional causality in mean from the stock price index to the exchange rate before the BR. The null hypothesis of no causality from lag 1 to lag 10 could not be rejected for other pairs

before the BR. After the BR (before the USE), no causality in mean was detected between the variables. After the USE, a unidirectional causality in mean from the Nikkei to the JPY and from the TOPIX to the Nikkei was detected. Furthermore, the cross-correlation coefficients values dramatically changed after each event. These findings mean that returns of stock price indexes have useful information for predicting exchange rate returns before the BR. After the USE, returns of the Nikkei price index have information useful to predicting returns of the JPY exchange rate, and returns of the TOPIX price index have information useful for predicting returns of the Nikkei price index.

**Table 3.** Causality in mean.

| Pair | Before BR | After BR (Before USE) | After USE |
|---|---|---|---|
| JPY to Nikkei | −0.0387 | 0.0190 | 0.0337 |
| Nikkei to JPY | 0.2852 *** | 0.0581 | 0.0749 * |
| JPY to TOPIX | −0.0871 | −0.0219 | 0.0547 |
| TOPIX to JPY | 0.2625 *** | −0.0669 | 0.0000 |
| Nikkei to TOPIX | −0.0029 | 0.1284 | −0.0423 |
| TOPIX to Nikkei | 0.0285 | −0.1518 | −0.0304 ** |

Note: This table reports the cross-correlation coefficients between the standardized residuals with the largest absolute value for Hong's (2001) *Q*-statistics within 10 lags. ***, ** and * mean to reject the null hypothesis of no causality from lag 1 to lag 10 at 1%, 5%, and 10% significance levels.

Table 4 presents the coefficients of the cross-correlation function with the maximum absolute value of Hong's (2001) *Q*-statistics within the first 10 lags for the squared standardized residuals. The coefficients showed bidirectional causality in variance between exchange rates and stock price indexes for the whole period and between stock price indexes after the BR and USE. This means volatility in one market has spread to other markets. Cross-correlation coefficient values and asymptotic powers dramatically change after each event.

**Table 4.** Causality in variances.

| Pair | Before BR | After BR (Before USE) | After USE |
|---|---|---|---|
| JPY to Nikkei | −0.0562 * | 0.0132 ** | −0.0032 ** |
| Nikkei to JPY | 0.4434 *** | −0.0478 ** | 0.6210 *** |
| JPY to TOPIX | −0.0629 * | −0.0114 ** | 0.0184 ** |
| TOPIX to JPY | 0.3480 *** | −0.0413 ** | 0.5774 *** |
| Nikkei to TOPIX | −0.0562 | −0.0072 ** | 0.0123 * |
| TOPIX to Nikkei | 0.1631 | −0.0141 ** | 0.0023 * |

Note: This table reports the cross-correlation coefficients between the squared standardized residuals with the largest absolute value for Hong's (2001) *Q*-statistics within 10 lags. ***, ** and * mean to reject the null hypothesis of no causality from lag 1 to lag 10 at 1%, 5%, and 10% significance levels.

## 5. Concluding Remarks

This paper examines the responses of exchange rates and stock price indexes in Japan to information and news about the BR and USE results. The empirical findings showed that exchange rates and stock price index returns were significantly affected by the BR and USE within two weeks of the events.

The BR caused exchange rates to appreciate and the stock price index (the TOPIX) to decrease. The BR increased the volatility of exchange rates (in the first week), causing instability in Japanese financial markets. This is in line with the findings of Belke et al. (2018), who demonstrated that BR-induced policy uncertainty caused instability in financial markets in the UK and other European countries.

The USE caused exchange rates to depreciate and the stock price index (the Nikkei) to increase. This finding is consistent with Shaikh (2017) argument about the effect of the USE on equity and foreign exchange markets across the globe.

Asymmetric impact coefficients were statistically significant for the variance of stock price indexes, demonstrating the stronger effect of negative news than positive news on the volatility of the Japanese stock market for the period under estimation.

The BR and USE affected the causality relationship between the exchange rate and stock price indexes. The cross-correlation coefficient values and the asymptotic power of Hong's (2001) *Q*-statistics dramatically changed after both events.

When information about the BR and USE results first spread, its impacts, to a certain degree, depended on the coverage and interpretations of events by professionals (e.g., scientists, broadcasting corporations, and information agencies) and nonprofessionals (mostly through social media). Therefore, appropriate interpretations of the events and rational predictions of the possible outcomes affecting the behavior and decisions of individuals and firms also have impacts on financial markets. This paper's findings contribute to the literature on the impact of information about international political and economic events on the JPY exchange rate and the Nikkei and TOPIX price indexes and present information useful for policymakers and investors interested in Japanese financial markets.

**Funding:** This research was supported by a grant-in-aid from the Japan Society for the Promotion of Science (JSPS, Grant-in-Aid for Scientific Research (C), Number 19K01756).

**Acknowledgments:** The author thanks the participants in the international economic policy session of the 18th International Conference of the Japan Economic Policy Association (JEPA), the regional financial issues session of the 94th Annual Conference of Western Economic Association International (WEAI), and the three anonymous referees for their helpful comments and suggestions. The author alone is responsible for any errors that may remain.

**Conflicts of Interest:** The author declares no conflict of interest.

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
