# Peer review of "The Impacts of International Political and Economic Events on Japanese Financial Markets"

_ijfs, doi:10.3390/ijfs8030043_

Round 1

Reviewer 1 Report

The updated version addresses properly all minor shortcomings of the first version.

Author Response

Thank you very much for your support in the improvement of the paper!

Reviewer 2 Report

I am satisfied with the changes and I believe the manuscript is ready for publication.

Author Response

Thank you very much for your support in the improvement of the paper!

Minor spell check was done. In particular, the section numbers were adjusted.

Reviewer 3 Report

I would like to start by congratulating the author for the effort put into refining the article. Despite this fact, after considering the resubmission in comparison to the original version I see that very little work has been conducted along the specified lines. 

1. Summary and overall opinion

The manuscript investigates the impact of news advent on financial markets, focusing on the case of Japan. The theoretical construction remains equally as poor as in the original. The simulation has not been significantly upgraded and still uses classic/mainstream investigation tools that however seem not to be soundly applied. The discussion is generally coherent though vague and the manuscript seems updated in terms of references. The manuscript has a limited dose of intrinsic merit and is still not publishable as it stands.

2. Comments and Suggestions

1. In the original review, I pointed to the fact that the research question is in my opinion rather thin and should be heavily discussed in the introduction which appears to be one of the weakest sections of the manuscript. Despite the fact that the author is successful in detecting a gap in the literature that paper aims to fill, the original contributions of the manuscript are missing. This is a problem that needs to be considered first given the fact that at first glance the actual contributions appear to be rather subtle. In addition to this, the section is in a state that requires heavy upgrades.

The author has done nothing in the direction of upgrading the introduction or to present original contributions. 

2. The methodological exercise remains a casual formulation and lacks consistency. In the previous report I have indicated ways of enriching the approach content-wise. Please see the comment below. 

There is an important vein of literature dealing with the impact of fresh information on financial markets. I advise the authors to reconsider their approach in order to gain complexity. The simplest exercise possible would be the incorporation of an event-study methodology based on a battery of specifications from the GARCH family with and without asymmetries. Useful examples here include GARCH, EGARCH, GARCH – GJR, ZARCH, A-PARCH, FIGARCH, NAGARCH, or IGARCH, among others. 

The authors could consider certain combinations as a baseline specification and then further test for robustness. Enriching the typical event study set-up would also be a bonus.

3. In review one, I have instructed the author to expand the sample as the current one is not relevant for the methodology employed, and therefore the results can't be considered as sound. No efforts have been to solve this.

4. Several changes have been performed in the results section. However, the author was unsuccessful in meeting the recommendations. 

5. The conclusions section is a bit better than the original but again fails to offer several clear take-aways that derive from the study.

6. I also advised the author to incorporate a discussion section to add consistency. This problem also remains unsolved.

Minor concerns:

  1. The language remains a serious problem. 
  2. The manuscript has 2 sections both labeled as Section 3. 

Author Response

Thank you very much for your support in the improvement of the paper!

Comment: "The manuscript has 2 sections both labeled as Section 3"

Reply: The section numbers were adjusted.

Note: We do not consider other comments appropriate, as we have already addressed them in the previously revised version of the paper.